# Biferroelectricity of a homochiral organic molecule in both solid crystal and liquid crystal phases

Xian-Jiang Song[1,2], Xiao-Gang Chen[1,2], Jun-Chao Liu[1,2], Qin Liu[1], Yi-Piao Zeng[1], Yuan-Yuan Tang [1], Peng-Fei Li [1], Ren-Gen Xiong [1]✉ & Wei-Qiang Liao [1]✉

Ferroelectricity, existing in either solid crystals or liquid crystals, gained widespread attention from science and industry for over a century. However, ferroelectricity has never been observed in both solid and liquid crystal phases of a material simultaneously. Inorganic ferroelectrics that dominate the market do not have liquid crystal phases because of their completely rigid structure caused by intrinsic chemical bonds. We report a ferroelectric homochiral cholesterol derivative, $\beta$-sitosteryl 4-iodocinnamate, where both solid and liquid crystal phases can exhibit the behavior of polarization switching as determined by polarization–voltage hysteresis loops and piezoresponse force microscopy measurements. The unique long molecular chain, sterol structure, and homochirality of $\beta$-sitosteryl 4-iodocinnamate molecules enable the formation of polar crystal structures with point group 2 in solid crystal phases, and promote the layered and helical structure in the liquid crystal phase with vertical polarization. Our findings demonstrate a compound that can show the biferroelectricity in both solid and liquid crystal phases, which would inspire further exploration of the interplay between solid and liquid crystal ferroelectric phases.

Ferroelectrics are electro-active materials that feature switchable spontaneous polarization, which has sparked great interest in their applications in nonvolatile memory elements, capacitors, sensors, micro-actuators, and optoelectronic devices[1–6]. The discovery of ferroelectricity in Rochelle salt opens up the history of ferroelectrics since 1920[7]. Ferroelectricity comes from the directional arrangement of electric dipoles, basically existing in the crystalline or semi-crystalline solid materials with polar symmetry, including inorganic crystals or ceramics[8–11], molecular crystals[2,12–26], and polymers[27,28], rather than fluid phases. Although the liquid crystal (LC) was discovered as early as 1888 by the botanist Friedrich Reinitzer[29], which has revolutionized the display industries[30–33], the first ferroelectric liquid crystal (FLC) p-decyloxybenzylidene p'-amino 2-methyl butyl cinnamate (DOBAMBC) was not discovered until 1975 by Meyer et al.[34].

Compared to solid ferroelectric materials, the symmetry considerations on ferroelectricity for most LCs are obviously a bit more complicated. The molecular arrangements in most common LC phases, such as nematic (N) and smectic (SmA, SmC, etc.) ones are inherently incompatible with the presence of spontaneous polarization. If the molecules in a tilted smetic phase are chiral (such as SmC*) and have an electric dipole moment perpendicular to the molecular long axis, such LCs could be ferroelectric, e.g., the well-studied DOBAMBC[34–38]. In the past decades, a variety of excellent FLCs have emerged that are expected to be used in the next-generation display and photonics devices[39–41]. However, ferroelectricity has never been observed simultaneously in both solid crystal (SC) and LC phases (biferroelectricity) of material so far.

Phase rule of Gibbs is a general principle that governs "pVT" systems in thermodynamic equilibrium (p = pressure, V = volume, and

[1]Ordered Matter Science Research Center, Nanchang University, 330031 Nanchang, People's Republic of China. [2]These authors contributed equally: Xian-Jiang Song, Xiao-Gang Chen, Jun-Chao Liu. ✉e-mail: xiongrg@seu.edu.cn; liaowq@ncu.edu.cn

$T$ = temperature), that is $F = C − P + 2$, where $F$ is the number of degrees of freedom, $C$ is the number of components, and $P$ is the number of phases[42]. When a solid crystal undergoes a ferroelectric structural phase transition, $\Delta F$ should be zero because $C$ and $P$ do not change in these two phases. But if a material undergoes a transition from a SC phase to a LC phase, then $\Delta F$ may not be zero because of the change of $P$ in these two phases. Therefore, if both SC and LC phases in a material can be ferroelectric, this breakthrough will revolutionize the field of ferroelectrics. Unfortunately, inorganic ferroelectrics that dominate the application market are almost impossible to achieve ferroelectric properties in both SC and LC phases due to their intrinsic chemical bonds.

Here, we present a homochiral cholesterol derivative, β-sitosteryl 4-iodocinnamate (4I-CASS). In the SC phases, 4I-CASS shows chiral and polar packing, while the tilted molecules are arranged in a vertical helix to create molecular dipoles in the LC phases. Intriguingly, the detected polarization-switching behaviors demonstrate that the ferroelectricity exists in the SC and LC phases of 4I-CASS simultaneously (Fig. 1). 4I-CASS is thus a compound that can exhibit biferroelectricity in both SC and LC phases. Organic molecules can form flexible units to induce the emergence of LC phases, which are unavailable in inorganic compounds. The homochirality of the tilted organic molecules not only promotes the possibility of crystallizing in the polar structure for SC phases but also makes the molecular orientation process around the layer normal line that might result in the oriented alignment of electric dipole moments to induce ferroelectricity for LC phases. These findings would inspire further exploration of organic ferroelectric SCs and LCs and the interplay between SC and LC ferroelectric phases.

## Results and discussion

Cholesteric molecules can be used as a structural unit to construct LC with some special helical pattern. Such pattern combines with the chirality of the molecular ends to constrain arbitrary rotations along the molecular long axis and generate the permanent dipole in the vertical orientation, providing a clue to the design of desired FLCs. The cholesterol derivatives β-sitosteryl 4-iodocinnamate (4I-CASS), cholesteryl 4-iodocinnamate (4I-CACS), and dihydrocholesteryl 4-iodocinnamate (4I-CAHCS) plate crystals were grown by the simple solution method with slow evaporation of respective ethyl acetate solutions at room temperature. Structural analyses reveal that all of them crystallize in the same polar monoclinic space group $P2_1$ at 300 K (Supplementary Table 1). The asymmetric unit of crystal structure in each compound consists of two crystallographically independent molecules (Fig. 2a and Supplementary Fig. 1). The 4I-CASS molecule has a different alkyl side chain from that of the 4I-CACS and 4I-CAHCS molecules. The steroid skeleton of 4I-CASS and 4I-CACS molecules shows a double bond, while that of 4I-CAHCS has no double bond. For

the packing view of the structure, the benzene rings of adjacent molecules in 4I-CASS are stacked parallelly along the $b$-axis, and the C−I bonds are almost perpendicular to the $b$-axis (Supplementary Fig. 2). In 4I-CACS and 4I-CAHCS, the benzene rings of adjacent molecules are parallel-stacked along the $a$-axis, while the C−I bonds are almost parallel to the $b$-axis (Supplementary Figs. 2–4). As Fig. 2c shows, the 4I-CASS molecules are each stacked by the symmetry operation of two-fold screw axes to form a structure similar to a double-row paddle. Above the structural phase transition temperature of 342 K for 4I-CASS, its crystal symmetry changes from $P2_1$ in phase I (the phase below the solid-to-solid phase transition temperature of 342 K) to $C2$ in phase II (the phase above the solid-to-solid phase transition temperature of 342 K while below the solid-liquid crystal phase transition temperature of 444 K), both of which have a similar packing structure, except for the doubling of the unit cell because of the presence of the two-fold rotation axes in phase II (Fig. 2c, e and Supplementary Fig. 5). As shown in the spatial symmetry operation change during the phase transition (Fig. 2d), the symmetry operation of two-fold rotation axes appears after transitioning from phase I to phase II. In phase II, the thermal motions of some carbon atoms in the 4I-CASS molecular tails are intensified to become disordered to a certain extent, so each of them is split to be modeled as two equivalent positions (Fig. 2b).

The dipole moment of a single molecule of 4I-CASS, 4I-CACS, and 4I-CAHCS were calculated, respectively. As shown in Supplementary Figs. 6–9, the dipole moments of these molecules are similar, all around 1.8 Debye (Supplementary Table 2). The directions of molecular dipoles are also very close, almost all of which are parallel to the direction of the carbon−oxygen single bond. From the packing view, the arrangements of these molecules make the macroscopic total dipole moments cancel each other in the [1 0 0] direction and [0 0 1] direction, respectively (Fig. 2c), leaving only along the [0 1 0] direction (Supplementary Fig. 5), which can be clearly observed from the slanted alignment of carbon−oxygen single bonds. This also meets the crystallographic symmetry requirements of point group 2, and thus the polarization of these crystals is all along the polar $b$-axis, while the polarization along $a$- and $c$-axis is zero. Furthermore, in order to estimate the ferroelectric polarization of the crystals, we employed Berry phase method to gain the total value of ferroelectric polarization, from which the polarization with 181.93, 233.17, 27.12, and 28.53 nC/cm² can be extracted for 4I-CASS crystal at 300 K, 4I-CASS crystal at 363 K, 4I-CACS crystal at 300 K and 4I-CAHCS crystal at 300 K, respectively (Supplementary Table 2). The difference in polarization value mainly comes from different molecular dipole arrangement patterns in each crystal. It should be noted that the directions of the molecular long axis of 4I-CASS in the lattice are toward the $a$-axis, while the directions of the molecular long axis of 4I-CACS and 4I-CAHCS are along the polar $b$-axis (Supplementary Figs. 1–5), which lead to the difficulty of polarization switching for the latter two. In contrast, since the molecular long axis of 4I-CASS is almost perpendicular to the [0 1 0] direction, a small twist of the molecules may be sufficient to enable the polarization reversal, inducing the desired ferroelectricity.

Differential scanning calorimetry (DSC) analysis reflects that 4I-CASS experiences a solid-to-solid structural phase transition at around 342 K (Fig. 3a), while 4I-CACS and 4I-CAHCS do not have structural phase transition behaviors. From DSC curves, all of 4I-CASS, 4I-CACS, and 4I-CAHCS experience a phase transition from solid crystal phase to liquid crystal phase with a transition temperature of 444, 442, and 423 K, respectively (Fig. 3a and Supplementary Fig. 10,11). Correspondingly, the real part ($\varepsilon'$) of the dielectric permittivity shows remarkable step-like anomaly near 342 K for 4I-CASS and λ-like anomalies at around 444, 442, and 423 K for 4I-CASS, 4I-CACS, and 4I-CAHCS (Supplementary Figs. 10–12), respectively. Particularly, the DSC curves show that 4I-CASS has two liquid crystal phases with a transition temperature of 431 K measured in the cooling run. In addition, the

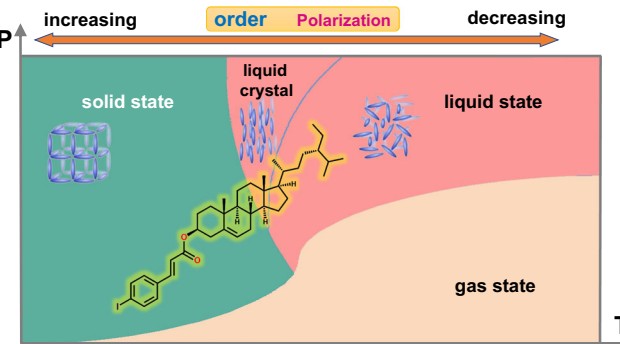

**Fig. 1 | Guiding ideology for obtaining the ferroelectric 4I-CASS.** 4I-CASS can show the biferroelectricity in both solid and liquid crystal phases. The light blue shapes represent the degree of molecular order in the solid, liquid crystal, and liquid phases. The inset molecule shows the molecular structure of 4I-CASS.

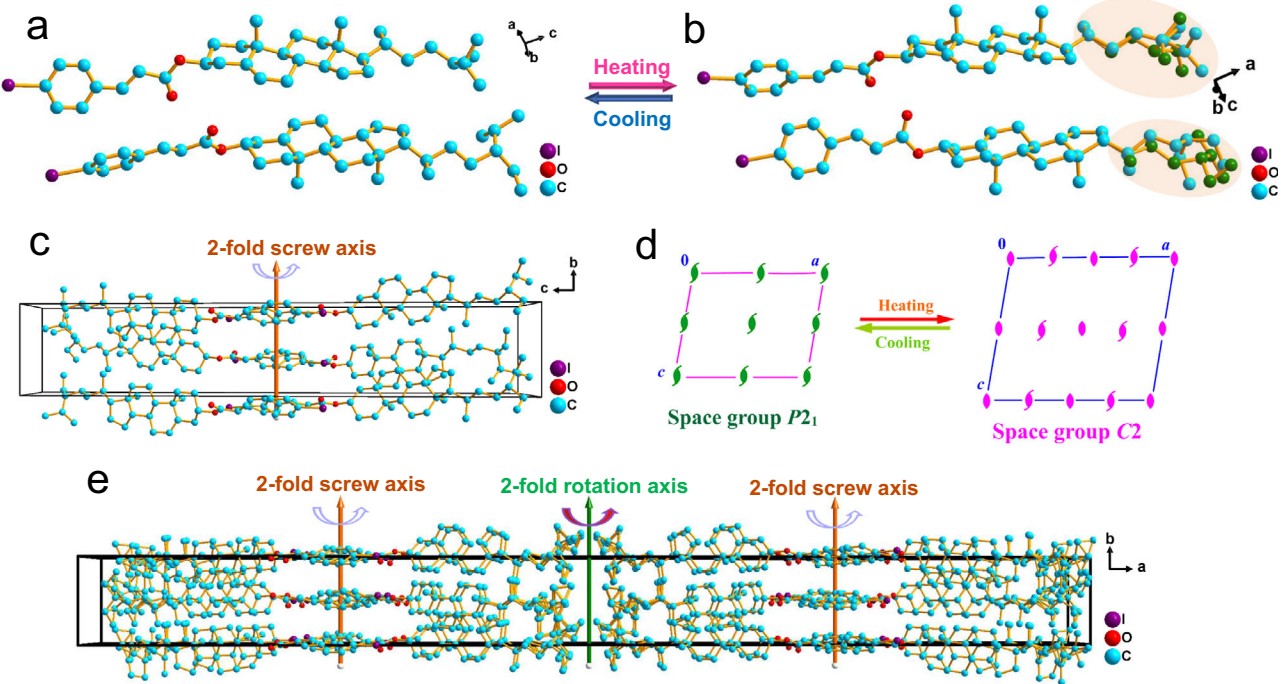

**Fig. 2 | Crystal structures of 4I-CASS. a, b** The asymmetric unit in phase I (**a**) and phase II (**b**), respectively. **c** Packing view in phase I along the [1 0 0] direction. **d** Diagram of the change in spatial symmetry operations from phase I to phase II. In phase I, the green symbols stand for the symmetry operation of two-fold screw axes. In phase II, the pink elliptical shape symbols denote the symmetry operation of two-fold rotation axes, while the other pink symbols represent the symmetry operation of two-fold screw axes. **e** Packing view in phase II along the [0 0 1] direction.

clearing point and crystallization temperature point of 4I-CASS are 490 and 389 K, respectively.

Normally, polarized light microscopy is a common means to determine the textures of liquid crystals. The liquid crystal phases of these compounds are formed under thermal induction. As the temperature rose to above 444 K, the sample of 4I-CASS softened and entered the LC state, by the observation under perpendicular polarized light (Fig. 3c). When the temperature increased to 453 K, the sample exhibited a typical oil streak-like texture of cholesteric liquid crystals. Subsequently, the visual field became dark with the temperature continuing to rise to 500 K, which means that the LC phase disappeared and entered an isotropic liquid state. During the cooling process, the fingerprint-like texture and focal-conic texture appeared, which is the presentation of the typical characteristic of cholesteric liquid crystals. When the directors of the cholesteric liquid crystal helical molecules are perpendicular, parallel, or oblique to the glass slide surface, respectively, oil streak, fingerprint, and focal-conic textures will appear correspondingly. However, as the temperature decreased to 423 K, the texture morphology further changed with the appearance of a broken lined focal-conic texture, meaning the appearance of a SmC* phase. The texture features of these LC phases are also manifested under the change of pressure (Supplementary Fig. 13). Similarly, the 4I-CACS and 4I-CAHCS also have cholesteric LC phases observed by their changes of oil streak-like to focal-conic textures under polarized light (Supplementary Figs. 14–16).

Moreover, to further accurately determine the transition between different phases of 4I-CASS, we performed variable temperature powder X-ray diffraction (PXRD) measurements. From Fig. 3b, with the increase in temperature, the PXRD pattern at 423 K shows some changes accompanied by the shift and splitting of several peaks compared to that at 300 K, corresponding to the transition of phase I to phase II. When the temperature increased to 463 K, most of the diffraction peaks in the PXRD pattern disappeared, indicating entry into the cholesteric liquid crystal phase. In the subsequent cooling

runs, a distinct diffraction peak of 5.5° appeared at 408 K, suggesting the formation of a LC phase with a molecular layer spacing of 16.04 Å. The ratio of the interlayer spacing to the molecular length (25.88 Å) is less than 1, indicating that this liquid crystal LC phase is SmC* phase. Upon cooling back to 300 K, the 4I-CAHCS reverted to a crystalline phase. It should be pointed out that the structural phase transition from phase I to phase II is completely reversible in the temperature range below the transition temperature of 444 K.

The molecular arrangement of 4I-CASS in the SmC* phase satisfies both the molecular layer structure and the helical structure of tilted chiral molecules, and the molecular long axes are parallel to each other in each plane that is perpendicular to the helical axis, resulting in the spontaneous polarization perpendicular to the molecular long axis. Generally, the length of the helical pitch $HP$, that is, the distance between layer A and layer B in Fig. 4a, is $-10^3 d$, where $d$ represents the spacing between adjacent layers[43]. Therefore, from a local perspective, in several adjacent layers, the molecular arrangement in each layer can be regarded as approximately the same, so it is conceivable that there is a two-fold axis parallel to the $y$-axis in the intermediate layer, as shown in the enlarged structure in the adjacent three layers in Fig. 4a. However, on the whole, the precession of the molecular orientation in each layer around the $z$-axis needs to be considered, and thus the two-fold axis disappears at this time, making the total net polarization zero. Under an applied electric field, the dipoles of each layer will be aligned along the direction of the electric field. In this case, the helical pitch of molecular precession tends to be infinite, and the spontaneous polarization of the layers will point in the same direction (Fig. 4b). To detect the ferroelectricity of 4I-CASS in the LC phase, we performed the polarization–voltage loop measurements. The reversal of the direction of spontaneous polarization can cause a characteristic current response for an applied triangular electric field (Supplementary Fig. 17), which can be used to determine the value of polarization[44]. As shown in Fig. 4c, the

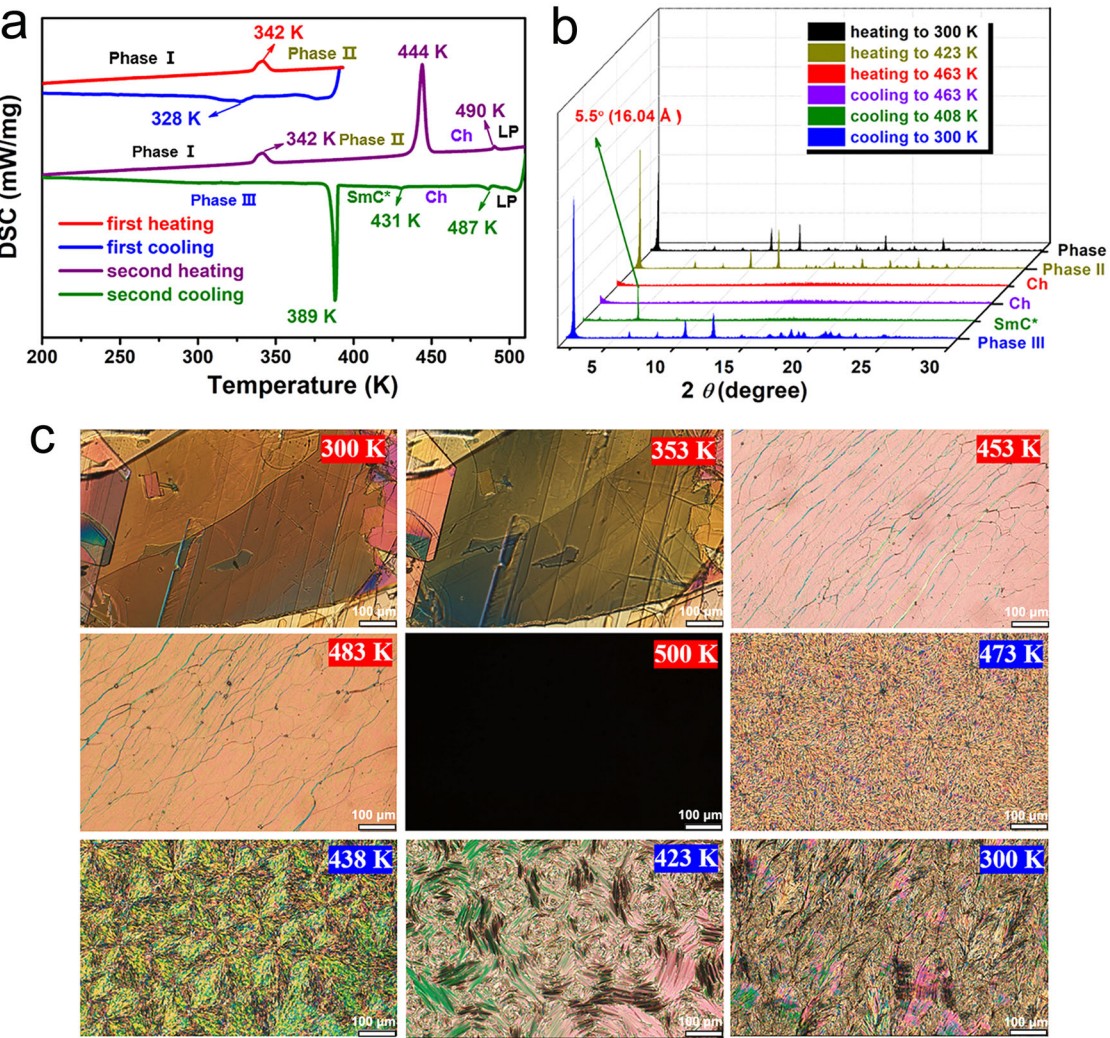

**Fig. 3 | Phase transitions of 4I-CASS. a** DSC curves of 4I-CASS. In the first heating and cooling runs, the phase below 342 K is phase I, the phase above 342 K but below 444 K is phase II, and the structural phase transition from phase I to phase II is completely reversible in the temperature range below the transition temperature of 444 K. In the second heating run, the phase between 444 and 490 K is the cholesteric LC phase (Ch), and the phase beyond 490 K is liquid phase (LP). In the second cooling run, an emerging LC phase between 389 and 431 K is SmC* phase, and a crystalline phase (phase III) is generated below 389 K. **b** Temperature-dependent PXRD patterns of 4I-CASS. **c** Polarized photomicrographs of 4I-CASS in phase I (300 K), phase II (353 K), Ch (453 K and 483 K), and LP (500 K) during the heating process, and Ch (473 K and 438 K), SmC* (423 K), and phase III (300 K) during the cooling process.

polarization–voltage hysteresis loops were measured at 433 K, affording solid evidence for the ferroelectricity in the SmC* phase of 4I-CASS.

For the SC phases of 4I-CASS, the experimental testing of the polar state has also been carried out by measuring the electromechanical response of the thin film samples by piezoresponse force microscopy (PFM) measurements, which has been confirmed as an effective method to study the stable and switchable polarization[45–49]. Figure 5a–c shows the topography of a representative as-prepared spin-coating thin film along with its corresponding lateral PFM amplitude and phase images at room temperature. Bright and dark regions in the amplitude image represent the strength of in-plane polarization, and the black lines correspond to the domain walls. Different color tones in the PFM phase image represent different polarization directions. The size of the domains is in the micrometer scale (Fig. 5a–c and Supplementary Fig. 18), which is comparable to the domains observed in some previously reported molecular ferroelectric thin films such as imidazolium perchlorate[50], 2-(hydroxymethyl)−2-nitro-1,3-propanediol[51] and (−)-camphanic acid[52], but larger than the domains in some conventional inorganic ferroelectric

thin films like Pb(Zr,Ti)O$_3$[53]. The observation of domain structures provides initial evidence of the existence of spontaneous polarization for 4I-CASS in phase I.

A natural next step in establishing the ferroelectricity of the 4I-CASS would be the investigation of its polarization response under an applied electric field. The good-shape ferroelectric hysteresis loop measured at room temperature illustrates the ferroelectricity of 4I-CASS in phase I (Fig. 4d). Highly localized electric field produced by the PFM tip can be used for polarization switching as well. Here, we study the domain-switching behavior on the thin films obtained by drop coating, as this method can obtain flat films with larger areas. Figure 5d–f shows that scanning the central region with the PFM tip under an applied voltage of −100 V results in domain switching, which is a strong indication of polarization switching under the electric field. The switched polarization state can be back switched as shown in Fig. 5g–i. Topographic mapping does not show any sample damage during the electric poling. In the same way, we also observed the domain structure (Supplementary Fig. 19a–c) and domain-switching behavior (Supplementary Fig. 19d–i) for 4I-CASS in phase II, respectively. On the basis of the obtained structural data and PFM

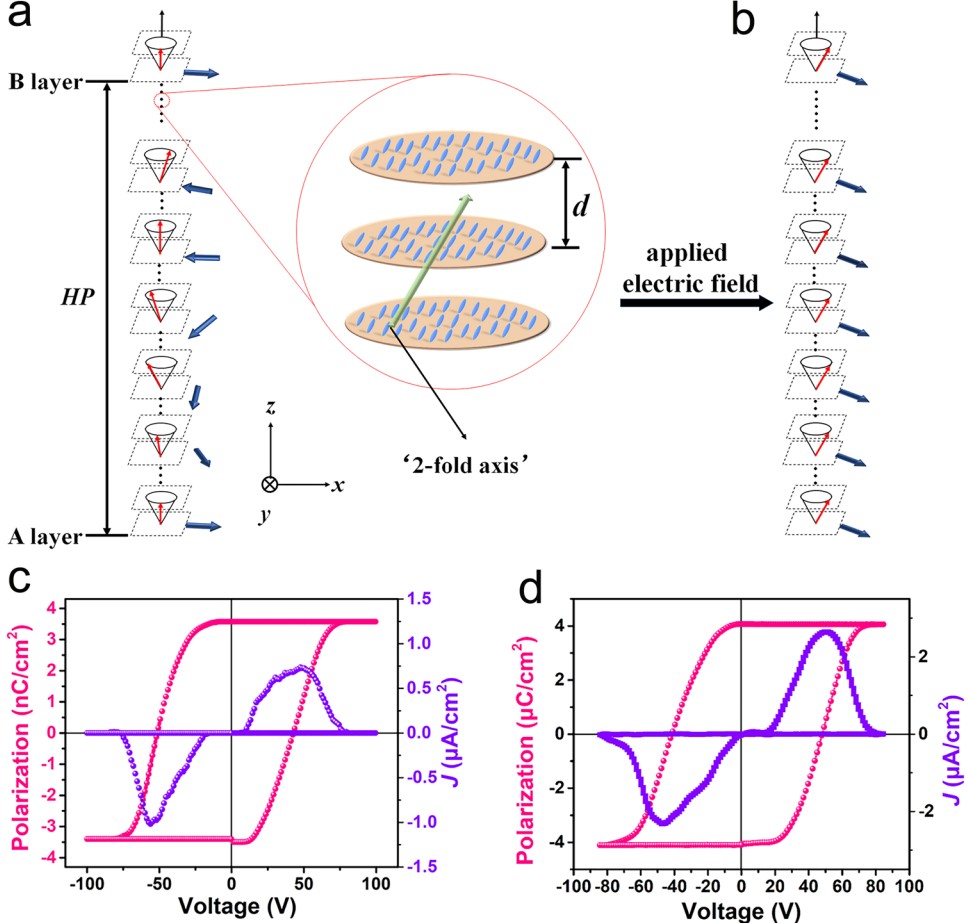

**Fig. 4 | Polarization switching for 4I-CASS. a** Scheme of the molecular arrangement in SmC* phase for 4I-CASS. The enlarged area of the red circle shows the molecular orientation between the adjacent three layers. The blue ellipses represent the 4I-CASS molecules, and the distance $d$ between the adjacent yellow disks represents the layer spacing. The red, blue, green, and black arrows represent the molecular orientation vector, layer polarization direction, the supposed two-fold axis and normal direction, respectively. The distance between layers A and B with same molecular orientation vector is the helix pitch (*HP*). **b** Scheme of spontaneous polarization under applied electric field. **c, d** Polarization–voltage loops in the SmC* phase (**c**) and phase I (**d**) for 4I-CASS, respectively.

characterization results, it is evident that 4I-CASS exhibits a switchable polar state in phase I and phase II, i.e., 4I-CASS is ferroelectric in both solid phases. We also studied the temperature-dependent evolution of the domain structures in SC phases. As shown in Supplementary Fig. 20, the domain patterns did not change significantly from phase I to phase II. This is due to the fact that the polarization directions of the crystal have not changed during the phase transition, as both phases belong to point group 2.

PFM switching spectroscopy measurements provide further evidence for the electric field-induced polarization switching in 4I-CASS. We recorded the PFM amplitude and phase response of the thin film sample in phases I and II using an AC voltage of 10 V. As shown in Supplementary Fig. 21a, b, typical butterfly amplitude curves with a 180° contrast in phase loops are observed in both phases due to the switching of polarization. We notice that the piezoelectric response (amplitude signal) of phase II is about 1.5 times larger than that of phase I (Supplementary Fig. 21a). This can be attributed to the larger polarization of the crystal in phase II. We obtained formants near the resonance frequencies of the two phases (Supplementary Fig. 21c). The curves are fitted well by the simple harmonic oscillator (SHO) model, from which the quality factor (Q factor) that describes the amplification of the signal can be obtained[54]. Here, the amplitude signal is corrected by dividing the Q factors. Furthermore, such measurements have been performed under gradually increasing excitation biases up to 10 V. We observed a linear relationship between the corrected

amplitude signal and the driving voltages, confirming that the response is piezoelectric (Supplementary Fig. 21d).

In summary, we demonstrate a ferroelectric chiral cholesterol derivative, 4I-CASS, which shows polarization-switching behaviors in both SC and LC phases. In solid crystal phases, the ferroelectricity is realized by the alignment of the molecules with the polar symmetry of point group 2, where the dipole moments are relatively easy to reorient due to being almost perpendicular to the molecular long axis, enabling switchable spontaneous polarization. The unique long-chain characteristics and chirality of 4I-CASS molecules facilitate the point symmetry of 2, and form the layered and helical structure in the cholesterol LC phase. The switchable spontaneous polarization perpendicular to the direction of molecular chains under the applied electric field was confirmed by $P - V$ hysteresis loops in LC phase. Solution-based processing crystal and thermally induced liquid crystal phase allow the deposition of 4I-CASS on various substrates, making it attractive for use in flexible electronics, biomedical devices, and other applications.

## Methods
### Materials
The 4-iodocinnamic acid was synthesized before the preparation of 4I-CASS, 4I-CACS, and 4I-CAHCS. The synthetic procedure of 4-iodocinnamic acid is shown in Supplementary Fig. 22. Malonic acid (3.12 g, 30.0 mmol) and piperidine (6 mmol) were added to a solution of 4-iodobenzaldehyde (4.64 g, 20.0 mmol) in pyridine (80 mL). The

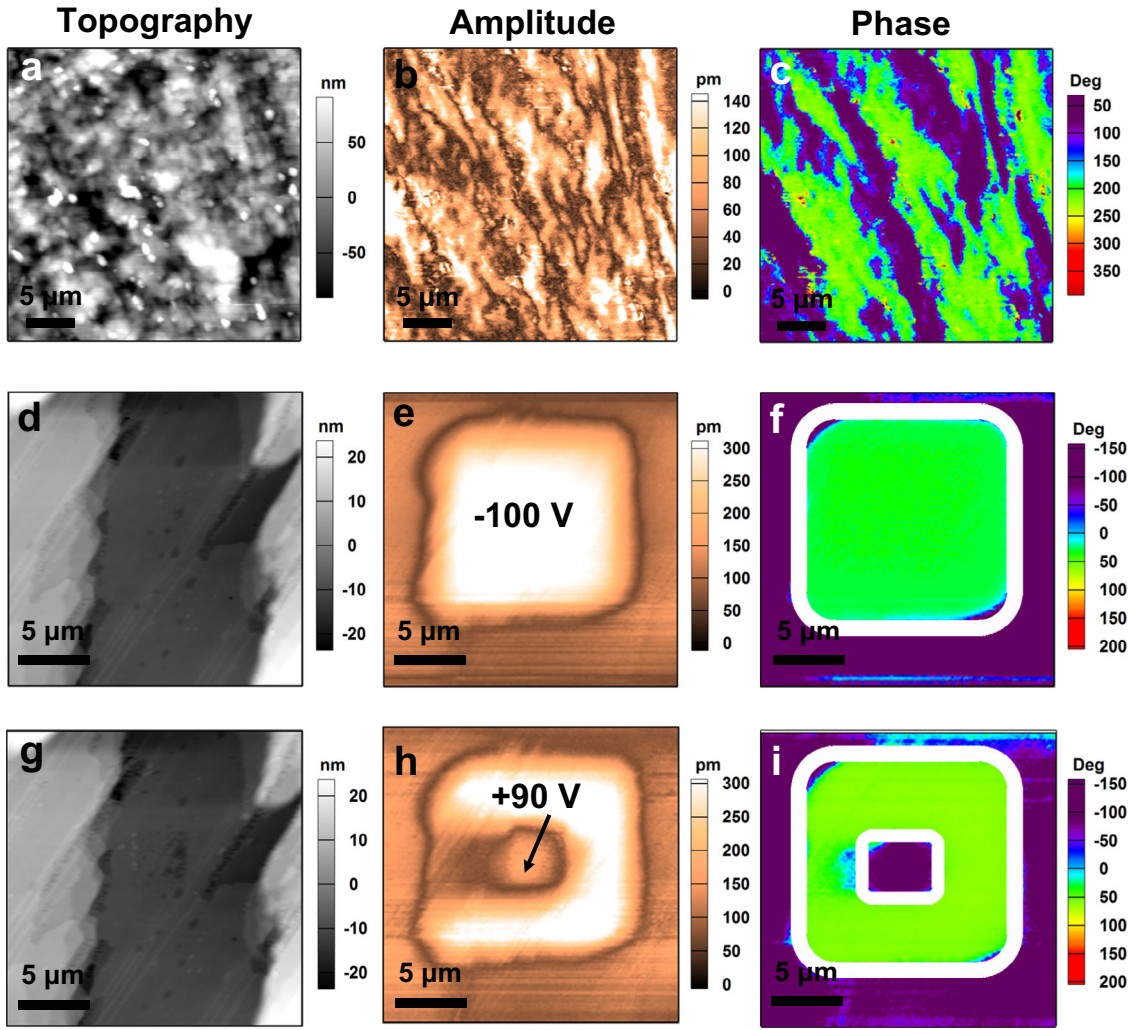

**Fig. 5 | PFM characterization of 4I-CASS in phase I. a–c** Topographic (**a**), lateral PFM amplitude (**b**), and phase (**c**) images mapped on the spin-coating thin film. **d–f** Topography (**d**), vertical PFM amplitude (**e**) and phase (**f**) images after applying a tip voltage of −100 V in the center area of a region with an initial state of a single domain state. **g–i** Topography (**g**), vertical PFM amplitude (**h**), and phase (**i**) images after applying +90 V tip voltage in the center region of the switched domain. The contrast of bright and dark in the topography images represents the height of the sample surface. The contrast of bright and dark in the amplitude images represents the magnitude of the piezoresponse. The violet and green regions in the phase images indicate the two different polarization-oriented states of ferroelectric domains. The white boxes indicate the area to which the tip voltage is applied.

reaction mixture was refluxed for 4 h. After the reaction was completed, the resultant mixture was acidified with 2 mol/L HCl to adjust the pH to 1–2 and then extracted with ethyl acetate (100 mL × 5). The organic layer was washed with deionized water, fully dried over with anhydrous $MgSO_4$, concentrated, and dried in a vacuum to give a white solid of 4-iodocinnamic acid.

For the synthesis of 4I-CASS (Supplementary Fig. 23), β-sitosterol (2.76 g, 5 mmol, 75%), 4-iodocinnamic acid (1.37 g, 5 mmol), DMAP (2.20 g, 18 mmol), and dicyclohexylcarbodiimide (3.09 g, 15 mmol) were added to a closed pressure-resistant bottle with 45 mL dichloromethane, and the mixture was stirred at 90 °C. After 48 hours, the resultant mixture was returned to room temperature, filtered, and washed with dichloromethane, and the solvent was removed under reduced pressure. The crude product was then purified by column chromatography (dichloromethane / petroleum ether = 1: 5) to obtain a white solid of 4I-CASS. The synthetic procedure of both 4I-CACS and 4I-CAHCS is similar to that of 4I-CASS and was described in the Supplementary Information (Supplementary Fig. 24 and Fig. 25). The crystals of 4I-CASS, 4I-CACS, and 4I-CAHCS were grown by slow evaporation of respective ethyl acetate solutions at room temperature.

### Measurements
Methods of single-crystal X-ray diffraction, powder X-ray diffraction, DSC and dielectric measurements, calculation condition, polarization–voltage loop measurements, and PFM measurements were described in the Supplementary Information.

## Data availability
All data generated and analyzed in this study are included in the Article and its Supplementary Information, and are also available from corresponding authors upon request. The crystal structures have been deposited in the Cambridge Crystallographic Data Centre under accession codes CCDC: 2189236-2189239, and can be obtained free of charge from the CCDC via www.ccdc.cam.ac.uk/data_request/cif.

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

## Acknowledgements

This work was supported by the National Natural Science Foundation of China (91856114 (W.-Q.L.), 21991142 (R.-G.X.), 21831004 (R.-G.X.), 22175082 (W.-Q.L.), and 21975114 (Y.-Y.T.)).

## Author contributions

R.-G.X. and W.-Q.L. designed and directed the studies. X.-J.S., X.-G.C., and J.-C.L. prepared the samples and characterized the properties. X.-G.C., Q.L., and Y.-P.Z. determined the structures. X.-J.S. and Y.-Y.T. did the PFM measurements. P.-F.L. performed the calculation. X.-J.S., X.-G.C., J.-C.L., R.-G.X., and W.-Q.L. wrote the manuscript. All the authors analyzed the data, discussed the results, and contributed to the manuscript.

## Competing interests

The authors declare no competing interests.
