## [Peer Review File · Nature Communications]

Biferroelectricity of a homochiral organic molecule in both solid crystal and liquid crystal phasesREVIEWER COMMENTS

Reviewer #1 (Remarks to the Author):

Molecular ferroelectrics have attracted intense pursuit in the scientific community in the past decades due to their light weight, easy preparation, and chemical modifiability. The ferroelectricity of reported molecule-based ferroelectric crystals and ferroelectric liquid crystals exists independently in the solid phase and the liquid crystal phase, respectively. So far, no single material has ferroelectricity in both solid and liquid crystal phases. Benefiting from the flexible structural tunability of the molecular system, the authors designed and synthesized a homochiral cholesterol derivative 4I-CASS, which experiences a solid-to-solid and a solid-to-liquid crystal phase transition and shows robust ferroelectricity in both solid and liquid crystal phases. Although the inorganic ferroelectrics dominate the application market, such biferroelectricity is not possibly achieved in them due to their rigid structure being inherently incompatible with the presence of the liquid crystal phase. I think this work is a breakthrough in the field of ferroelectrics, which will inspire further exploration of molecular ferroelectrics with biferroelectricity. Accordingly, I recommend publishing this work in nature communications as soon as possible.

Some minor comments:

1. The manuscript presents the different crystal structures of three cholesterol derivatives 4I-CASS, 4I-CACS, and 4I-CAHCS with the same P21 space group at 300 K, while the differences are not clearly shown and discussed.
2. In Figure 3, the scale bar is barely visible.
3. The piezoelectric response of phase II is larger than that of phase I. A corresponding explanation is required.
4. It is necessary to describe the synthesis details of β -sitosteryl 4-iodocinnamate and dihydrocholesteryl 4-iodocinnamate, although the synthetic procedures are similar to those of cholesteryl 4-iodocinnamate.

Reviewer #2 (Remarks to the Author):

As electro-active bistable materials, ferroelectrics have important applications in many technological fields, keeping them at the forefront of materials science, condensed matter physics, and chemistry. However, since the discovery of the first ferroelectric Rochelle salt in 1920, the research on ferroelectricity has been limited to a single solid crystal phase or liquid crystal phase, both theoretically and experimentally. Unprecedentedly, this work reports an interesting discovery of a homochiral cholesterol derivative with ferroelectricity in both the solid phase and liquid crystal phase. This work opens a new perspective for the study of ferroelectric materials. The experimental characterizations were designed comprehensively and carefully carried out. The result discussion is convincing and the conclusion is solid. I believe this work merits publication in Nature Communications without any doubt. A few minor points described below should be considered before publication.

1. Phases I and II need to be defined when they first appear in the main text.
2. In order to better show the structural differences of phases I and II, the two independent molecules in Fig. 2a and 2b should correspond to each other. Also in Fig. 2, Fig. 2d is not described in the main text.
3. For the polarization switching of 4I-CASS in the LC phase, the description of 'From a local

perspective (observing a few layers) in the tilted chiral molecules in Fig. 4a, the 2-fold axis is parallel to the layer where it is located. On the whole, however, due to the helical distribution of the molecules around the layer normal, the 2-fold axes do not actually exist, and thus the overall total net polarization is zero.' is confusing. Could the authors clearly discuss the 2-fold axis mentioned in this part?

4. In the last paragraph of the discussion part, the authors should give more information about the SHO model and Q factors.

5. The colors of atoms and bonds in Supplementary Fig. 1-3 are indistinguishable.

Reviewer #3 (Remarks to the Author):

The authors of this paper have succeeded in developing ferroelectricity in the chiral cholesterol derivative 4I-CASS in both crystal and liquid crystal phases (Fig. 1). Detailed crystal structure analysis of 4I-CASS (Fig. 2) revealed that the polar axis is the b axis. Furthermore, the polarization value is calculated by the Berry phase approach. Furthermore, by measuring DSC and polarized optical microscopy images, the phase transition in the liquid crystal phase is clarified (Fig. 3). In the SmC* phase, the ferroelectricity was confirmed by measuring the voltage dependence of the polarization value (polarization switching) (Fig. 4), and the polarization domain was successfully observed by PFM (Fig. 5). The present study is significant because 4I-CASS can be deposited on a variety of substrates and is expected to be applied to flexible electronics and other fields.

The main message by present experiments is clear, as Xian-Jiang Song et al. confirmed the ferroelectricity in solid and liquid phases in the same compound. I believe that the present finding is an important step for the polarization switching in liquid crystals, and highlights the importance of the ferroelectricity. However, I have questions and resultant comments to further improve the manuscript.

Comments.

1. Please insert the size scale bar in Fig. 5.
2. In Fig. 5, the domain image at zero electric field by PFM is reported. Please include data comparing the size of the domain to that of other compounds (normal solid or liquid) and how it differs from the domain of other compounds.
3. The measured spontaneous polarization value is about 4 nC/cm² (Fig. 4), please discuss how it differs from the polarization value estimated by the Berry phase method. In particular, is there the contribution of electronic polarization? In addition, if you can compare the retentivity with your calculations, please show us the results.
4. How do the polarized light microscope image in Fig. 3 and the PFM image in Fig. 5 compare in the same region? Of course, it depends on the temperature, but I think we can see how the domain changes with temperature.

REVIEWER COMMENTS

Reviewer #1 (Remarks to the Author):

Molecular ferroelectrics have attracted intense pursuit in the scientific community in the past decades due to their light weight, easy preparation, and chemical modifiability. The ferroelectricity of reported molecule-based ferroelectric crystals and ferroelectric liquid crystals exists independently in the solid phase and the liquid crystal phase, respectively. So far, no single material has ferroelectricity in both solid and liquid crystal phases. Benefiting from the flexible structural tunability of the molecular system, the authors designed and synthesized a homochiral cholesterol derivative 4I-CASS, which experiences a solid-to-solid and a solid-to-liquid crystal phase transition and shows robust ferroelectricity in both solid and liquid crystal phases. Although the inorganic ferroelectrics dominate the application market, such biferroelectricity is not possibly achieved in them due to their rigid structure being inherently incompatible with the presence of the liquid crystal phase. I think this work is a breakthrough in the field of ferroelectrics, which will inspire further exploration of molecular ferroelectrics with biferroelectricity. Accordingly, I recommend publishing this work in nature communications as soon as possible.

Response: We sincerely thank Reviewer 1 for his/her positive and valuable comments on our manuscript.

Some minor comments:

1. The manuscript presents the different crystal structures of three cholesterol derivatives 4I-CASS, 4I-CACS, and 4I-CAHCS with the same P21 space group at 300 K, while the differences are not clearly shown and discussed.

Response: Thank you for your valuable comment. According to your comment, we compared the crystal structures of 4I-CASS, 4I-CACS, and 4I-CAHCS in supplementary Fig. 1 and 2 in the revised manuscript to clearly show their structural

differences. As shown in supplementary Fig. 1 in revision, the basic unit of each compound contains two crystallographically independent molecules. The 4I-CASS molecule has a different alkyl side chain from that of the 4I-CACS and 4I-CAHCS molecules. The steroid skeleton of 4I-CASS and 4I-CACS molecules shows a double bond, while that of 4I-CAHCS has no double bond. For the packing view of the structure, the benzene rings of adjacent molecules in 4I-CASS are stacked parallelly along the *b*-axis and the C–I bonds are almost perpendicular to the *b*-axis (Supplementary Fig. 2a in revision). However, in 4I-CACS and 4I-CAHCS, the benzene rings of adjacent molecules are parallel-stacked along the *a*-axis, while the C–I bonds are almost parallel to the *b*-axis (Supplementary Fig. 2b-c). We have added the supplementary Fig. 1 and 2 and the corresponding discussion in the revised manuscript.

The corresponding changes in the revised main text:

“The asymmetric unit of crystal structure in each compound consists of two crystallographically independent molecules (Fig. 2a and Supplementary Fig. 1). The 4I-CASS molecule has a different alkyl side chain from that of the 4I-CACS and 4I-CAHCS molecules. The steroid skeleton of 4I-CASS and 4I-CACS molecules shows a double bond, while that of 4I-CAHCS has no double bond. For the packing view of the structure, the benzene rings of adjacent molecules in 4I-CASS are stacked parallelly along the *b*-axis and the C–I bonds are almost perpendicular to the *b*-axis (Supplementary Fig. 2). In 4I-CACS and 4I-CAHCS, the benzene rings of adjacent molecules are parallel-stacked along the *a*-axis, while the C–I bonds are almost parallel to the *b*-axis (Supplementary Fig. 2-4). As Fig. 2c shows, the 4I-CASS molecules are each stacked by the symmetry operation of 2-fold screw axes to form a structure similar to a double row paddle.”

The corresponding changes in the revised supplementary information:

Supplementary Fig. 1 (in revision) | Comparison of the molecular structure and the asymmetric unit of the crystal structure of 4I-CASS (a and b), 4I-CACS (c and d), and 4I-CAHCS (e and f) at 300 K.

Supplementary Fig. 2 (in revision) | Comparison of the packing view of the crystal structure of 4I-CASS (a), 4I-CACS (b), and 4I-CAHCS (c) at 300 K. The two different colors stand for two kinds of crystallographically independent molecules.

2. In Figure 3, the scale bar is barely visible.

Response: Thank you for your good comment. We have updated Figure 3 to make the scale bar of Figure 3c clear. Please see the revised Figure 3c below. According to your comment, we have also revised the previous supplementary Figure 11-14 (supplementary Fig. 13-16 in revision) to clearly show the scale bar.

The corresponding changes in the revised main text:

Fig. 3c (in revision) | c, Polarized photomicrographs of 4I-CASS in the heating and cooling runs.

3. The piezoelectric response of phase II is larger than that of phase I. A corresponding explanation is required.

Response: Thank you for this valuable comment. The polarization values of phase I and phase II calculated by the Berry phase method are 181.93 and 233.17 nC/cm², respectively. According to the definition of the piezoelectric constant, which characterizes the magnitude of the piezoelectric response, $d_{ij} = P_i/\sigma_j$, where P_i is the polarization and σ_j is the applied stress, the piezoelectric response is proportional to the polarization (*Phys. Rev. Lett.* 2020, 125, 207601). A larger spontaneous polarization in the phase II results in a larger piezoelectric response. We have added this explanation in the revised manuscript.

The corresponding changes in the revised main text:

“We notice that the piezoelectric response (amplitude signal) of phase II is about 1.5 times larger than that of phase I (Supplementary Fig. 21a). This can be attributed to the larger polarization of the crystal in phase II.”

4. It is necessary to describe the synthesis details of β -sitosteryl 4-iodocinnamate and dihydrocholesteryl 4-iodocinnamate, although the synthetic procedures are similar to those of cholesteryl 4-iodocinnamate.

Response: Thank you for your valuable comment. According to your comment, we have described the synthetic procedures of β -sitosteryl 4-iodocinnamate and dihydrocholesteryl 4-iodocinnamate in detail in the revised supplementary information (see below).

The corresponding changes in the revised supplementary information:

Scheme S3. Synthesis of β -sitosteryl 4-iodocinnamate.

Synthesis of β -sitosteryl 4-iodocinnamate (4I-CASS).

β -Sitosterol (2.76 g, 5 mmol, 75%), 4-iodocinnamic acid (1.37 g, 5 mmol), DMAP (2.20 g, 18 mmol), and DCC (3.09 g, 15 mmol) were added to a closed pressure-resistant bottle with 45 mL DCM, and the mixture was stirred at 90 °C. After 48 hours, the resultant mixture was returned to room temperature, filtered, washed with DCM, and the solvent was removed under reduced pressure. The crude product was then purified by column chromatography (DCM / PE = 1: 5) to obtain a white solid.

Scheme S4. Synthesis of dihydrocholesteryl 4-iodocinnamate.

Synthesis of dihydrocholesteryl 4-iodocinnamate (**4I-CAHCS**).

Dihydrocholesterol (1.94 g, 5 mmol), 4-iodocinnamic acid (1.37 g, 5 mmol), DMAP (2.20 g, 18 mmol), and DCC (3.09 g, 15 mmol) were added to a closed pressure-resistant bottle with 45 mL DCM, and the mixture was stirred at 90 °C for 48 hours. Then, the resultant mixture was cooled, filtered, washed with DCM, and the solvent was removed under reduced pressure. The crude product was then purified by column chromatography (DCM / PE = 1: 5) to obtain a white solid.

Reviewer #2 (Remarks to the Author):

As electro-active bistable materials, ferroelectrics have important applications in many technological fields, keeping them at the forefront of materials science, condensed matter physics, and chemistry. However, since the discovery of the first ferroelectric Rochelle salt in 1920, the research on ferroelectricity has been limited to a single solid crystal phase or liquid crystal phase, both theoretically and experimentally. Unprecedentedly, this work reports an interesting discovery of a homochiral cholesterol derivative with ferroelectricity in both the solid phase and liquid crystal phase. This work opens a new perspective for the study of ferroelectric materials. The experimental characterizations were designed comprehensively and carefully carried out. The result discussion is convincing and the conclusion is solid. I believe this work merits publication in Nature Communications without any doubt. A few minor points described below should be considered before publication.

Response: We sincerely thank Reviewer 2 for the valuable comments and his/her compliments on the impact and significance of our work.

1. Phases I and II need to be defined when they first appear in the main text.

Response: Thank you for your good comment. Phase I is the phase below the solid-to-solid phase transition temperature of 342 K. Phase II is the phase above the solid-to-solid phase transition temperature of 342 K while below the solid-liquid crystal phase transition temperature of 444 K. According to your comment, we have given a definition of phases I and II when they first appear in the revised manuscript.

The corresponding changes in the revised main text:

“Above the structural phase transition temperature of 342 K for 4I-CASS, its crystal symmetry changes from $P2_1$ in phase I (the phase below the solid-to-solid phase transition temperature of 342 K) to $C2$ in phase II (the phase above the solid-to-solid phase transition temperature of 342 K while below the solid-liquid crystal phase transition temperature of 444 K), both of which have a similar packing structure, except

for the doubling of the unit cell because of the presence of the 2-fold rotation axes in phase II (Fig. 2c,e and Supplementary Fig. 5).”

2. In order to better show the structural differences of phases I and II, the two independent molecules in Fig. 2a and 2b should correspond to each other. Also in Fig. 2, Fig. 2d is not described in the main text.

Response: Thank you for this good comment. We have revised Fig. 2a and 2b to make the two independent 4I-CASS molecules in phase I correspond to those in phase II. Please see the updated Fig. 2a and 2b below. Fig 2d shows that phase I has the spatial symmetry operation of 2-fold screw axes, while in phase II, besides the spatial symmetry operation of 2-fold screw axes, the spatial symmetry operation of 2-fold rotation axes also appears. According to your comment, we have added the corresponding description of Fig. 2d in the revised manuscript.

The corresponding changes in the revised main text:

Fig. 2a and 2b (in revision) | a,b, The asymmetric unit in phase I (a) and phase II (b), respectively.

“As shown in the spatial symmetry operation change during the phase transition (Fig. 2d), the symmetry operation of 2-fold rotation axes appears after transition from phase I to phase II.”

3. For the polarization switching of 4I-CASS in the LC phase, the description of ‘From a local perspective (observing a few layers) in the tilted chiral molecules in Fig. 4a, the 2-fold axis is parallel to the layer where it is located. On the whole, however, due to the helical distribution of the molecules around the layer normal, the 2-fold axes do not

actually exist, and thus the overall total net polarization is zero.' is confusing. Could the authors clearly discuss the 2-fold axis mentioned in this part?

Response: Thank you for your good comment. For easier understanding, we have made a new schematic diagram of the molecular arrangement of 4I-CASS in the SmC* phase. As shown in Fig. 4 (in revision), the chiral molecules are generally parallel to each other in each layer. By the way, due to the chirality of the molecules, the molecular orientation vector n of each layer rotates spirally around the layer normal (z -axis), and the spatial trajectory of n forms a cone surface (Fig. 4a). Generally, the length of the helical pitch HP , that is, the distance between layer A and layer B in Fig. 4a (in revision), is approximately 10^3d , where d represents the spacing between adjacent layers (*Mol. Cryst. Liq. Cryst.* 1988, 158, 1-150). Therefore, from a local perspective, in several adjacent layers, the molecular arrangement in each layer can be regarded as approximately the same, so it is conceivable that there is a 2-fold axis parallel to the y -axis in the intermediate layer, as shown in the enlarged structure in the adjacent three layers in Fig. 4a (in revision). However, on the whole, the precession of the molecular orientation in each layer around the z -axis needs to be considered, and thus the 2-fold axis disappears at this time. According to your comment, we have added the corresponding figures and discussion in the revised manuscript.

The corresponding changes in the revised main text:

“Generally, the length of the helical pitch HP , that is, the distance between layer A and layer B in Fig. 4a, is approximately 10^3d , where d represents the spacing between adjacent layers.⁴⁴ Therefore, from a local perspective, in several adjacent layers, the molecular arrangement in each layer can be regarded as approximately the same, so it is conceivable that there is a 2-fold axis parallel to the y -axis in the intermediate layer, as shown in the enlarged structure in the adjacent three layers in Fig. 4a. However, on the whole, the precession of the molecular orientation in each layer around the z -axis needs to be considered, and thus the 2-fold axis disappears at this time, making the total net polarization zero. Under an applied electric field, the dipoles of each layer will be aligned along the direction of the electric field. In this case, the helical pitch of

molecular precession tends to be infinite, and the spontaneous polarization of the layers will point into the same direction (Fig. 4b).”

Meanwhile, the related references have also been added in the revised manuscript:

44. Beresnev, L. A. *et al.* Ferroelectric Liquid Crystals. *Mol. Cryst. Liq. Cryst.*, **158**, 1-150 (1988).

Fig. 4 | Polarization switching for 4I-CASS. **a**, Scheme of the molecular arrangement in SmC* phase for 4I-CASS. The red, blue and green arrows represent the molecular orientation vector, layer polarization direction and the supposed 2-fold axis, respectively. **b**, Scheme of spontaneous polarization under applied electric field. **c,d**, Polarization–voltage loops in the SmC* phase (**c**) and phase I (**d**) for 4I-CASS, respectively.

4. In the last paragraph of the discussion part, the authors should give more information about the SHO model and Q factors.

Response: Thank you for your valuable comment. As can be seen from the methodology of the PFM characterization (Supplementary Information), we performed the PFM measurements at contact resonance to enhance the signal-to-noise ratio. In the vicinity of resonance, the amplitude and phase frequency response can be described using the simple harmonic oscillator (SHO) model:

$$A(\omega) = \frac{A_{\max}\omega_0^2/Q}{\sqrt{(\omega_0^2 - \omega^2)^2 + (\omega_0\omega/Q)^2}}$$

$$\tan \varphi(\omega) = \frac{\omega_0\omega}{Q(\omega_0^2 - \omega^2)}$$

where, A_{\max} is the amplitude at the resonance ω_0 , and Q the quality factor that describes energy losses in the system (*J. Am. Chem. Soc.* 2020, *142*, 20208-20215). The cantilever response at resonance is essentially multiplied by the quality factor (Q) of the cantilever. We have added information about the SHO model and Q factor in the revised manuscript and cited relevant literature.

The corresponding changes in the revised main text:

“The curves are fitted well by the simple harmonic oscillator (SHO) model, from which the quality factor (Q factor) that describes the amplification of the signal can be obtained.⁵⁵ Here, the amplitude signal is corrected by dividing the Q factors.”

Meanwhile, the related references have also been added in the revised manuscript:

55. Zhang, H.-Y. *et al.* Two-Dimensional Hybrid Perovskite Ferroelectric Induced by Perfluorinated Substitution. *J. Am. Chem. Soc.* **142**, 20208-20215 (2020).

5. The colors of atoms and bonds in Supplementary Fig. 1-3 are indistinguishable.

Response: Thank you for your careful reading. We have used different colors in the revised Supplementary Fig. 1-3 (Supplementary Fig. 3-5 in revision) to clearly

distinguish the colors of atoms and bonds. Please see the updated Supplementary Fig. 1-3 (Supplementary Fig. 3-5 in revision) below.

The corresponding changes in the revised supplementary information:

Supplementary Fig. 3 (in revision) | a,b, Packing view of crystal structures of 4I-CACS along the [1 0 0] direction (a) and the [0 0 1] direction (b), respectively.

Supplementary Fig. 4 (in revision) | a,b, Packing view of crystal structures of 4I-CAHCS along the [1 0 0] direction (a) and the [0 0 1] direction (b), respectively.

Supplementary Fig. 5 (in revision) | a,b, Packing view of crystal structures of 4I-CASS in phase I (**a**) and phase II (**b**) along the [0 1 0] direction, respectively.

Reviewer #3 (Remarks to the Author):

The authors of this paper have succeeded in developing ferroelectricity in the chiral cholesterol derivative 4I-CASS in both crystal and liquid crystal phases (Fig. 1). Detailed crystal structure analysis of 4I-CASS (Fig. 2) revealed that the polar axis is the b axis. Furthermore, the polarization value is calculated by the Berry phase approach. Furthermore, by measuring DSC and polarized optical microscopy images, the phase transition in the liquid crystal phase is clarified (Fig. 3). In the SmC* phase, the ferroelectricity was confirmed by measuring the voltage dependence of the polarization value (polarization switching) (Fig. 4), and the polarization domain was successfully observed by PFM (Fig. 5). The present study is significant because 4I-CASS can be deposited on a variety of substrates and is expected to be applied to flexible electronics and other fields.

The main message by present experiments is clear, as Xian-Jiang Song et al. confirmed the ferroelectricity in solid and liquid phases in the same compound. I believe that the present finding is an important step for the polarization switching in liquid crystals, and highlights the importance of the ferroelectricity. However, I have questions and resultant comments to further improve the manuscript.

Response: We sincerely thank Reviewer 3 for the excellent and valuable comments on our manuscript.

Comments.

1. Please insert the size scale bar in Fig. 5.

Response: Thank you for your good comment. We have inserted the size scale bar in the revised Fig. 5. Please see the updated Fig. 5 below.

The corresponding changes in the revised main text:

Fig. 5 (in revision) | PFM characterization of 4I-CASS in phase I. a-c, Topographic (a), lateral PFM amplitude (b) and phase (c) images mapped on the spin-coating thin film. d-f, Topography (d), vertical PFM amplitude (e) and phase (f) images after applying a tip voltage of -100 V in the center area of a region with an initial state of a single domain state. g-i, Topography (g), vertical PFM amplitude (h) and phase (i) images after applying +90 V tip voltage in the center region of the switched domain.

2. In Fig. 5, the domain image at zero electric field by PFM is reported. Please include data comparing the size of the domain to that of other compounds (normal solid or liquid) and how it differs from the domain of other compounds.

Response: Thank you for your constructive comment. The size of domains in ferroelectric thin films is affected by morphology, grain boundaries, strain, defects, etc. Therefore, the domains in different regions of the same film may have various sizes. As can be seen from Figure 5 and the supplementary Fig. 18 (in revision) below, we observed domains with different sizes, but most of them are larger than 1 μm. This is comparable to the size of domains observed in previously reported molecular ferroelectric thin films, such as imidazolium perchlorate (*Angew. Chem. Int. Ed.* 2014,

53, 5064-5068), 2-(hydroxymethyl)-2-nitro-1,3-propanediol (*J. Am. Chem. Soc.* 2020, **142**, 13989-13995) and (–)-camphanic acid (*Chem. Sci.* 2022,**13**, 748-753), but larger than domains in conventional inorganic ferroelectric thin films, such as Pb(Zr,Ti)O₃ (*Appl. Phys. Lett.* 2015, **107**, 142903), BiFeO₃ (*ACS Appl. Mater. Interfaces* 2018, **10**, 11768-11775) and CuInP₂Se₆ (*Nat. commun.* 2020, **11**, 3623). Generally, the domain size in molecular ferroelectric films (micron scale) is larger than that in inorganic ferroelectric films (nano scale). This is attributed to the relatively larger formation energy of the domain walls in molecular ferroelectric films. In inorganic ferroelectrics (displacive type), the atoms at the domain walls are in equilibrium positions, leading to a flat polarization profile across the domain walls (*Phys. Rev. B* 2002, **65**, 104111). Thus, the domain walls are broad and their formation energy are small. However, the ferroelectricity of molecular ferroelectrics (order-disorder type) comes from molecular dipoles. It has no equilibrium state that the spontaneous polarization is zero, which leads the polarization changes abruptly across the domain walls, causing a larger formation energy. According to your comment, we have added the corresponding figure and discussion in the revised manuscript.

The corresponding changes in the revised main text:

“The size of the domains is in the micrometer scale (Fig. 5a-c and Supplementary Fig. 18), which is comparable to the domains observed in some previously reported molecular ferroelectric thin films such as imidazolium perchlorate,⁵¹ 2-(hydroxymethyl)-2-nitro-1,3-propanediol ⁵² and (–)-camphanic acid,⁵³ but larger than the domains in some conventional inorganic ferroelectric thin films like Pb(Zr,Ti)O₃.⁵⁴”

Meanwhile, the related references have also been added in the revised manuscript:

51. Zhang, Y. *et al.* A Molecular Ferroelectric Thin Film of Imidazolium Perchlorate That Shows Superior Electromechanical Coupling. *Angew. Chem. Int. Ed.* **53**, 5064-5068 (2014).

52. Ai, Y. *et al.* Six-Fold Vertices in a Single-Component Organic Ferroelectric with Most Equivalent Polarization Directions. *J. Am. Chem. Soc.* **142**, 13989-13995 (2020).

53. Ai, Y. *et al.* An organic plastic ferroelectric with high Curie point. *Chem. Sci.* **13**,

748-753 (2022).

54. Mtebwa, M. *et al.* Room temperature concurrent formation of ultra-dense arrays of ferroelectric domain walls. *Appl. Phys. Lett.* **107**, 142903 (2015).

The corresponding changes in the revised supplementary information:

Supplementary Fig. 18 (in revision) | Domain structures observed in different regions of the same film of 4I-CASS.

3. The measured spontaneous polarization value is about 4 nC/cm² (Fig. 4), please discuss how it differs from the polarization value estimated by the Berry phase method. In particular, is there the contribution of electronic polarization? In addition, if you can compare the retentivity with your calculations, please show us the results.

Response: Thank you for this valuable comment. The calculated value of ferroelectric polarization (181.93 nC/cm²) of 4I-CASS is based on the *solid crystal* phase. On the other hand, the measured spontaneous polarization value is about 4 nC/cm² in *liquid crystal* phase. The ferroelectric polarization value of the *liquid crystal* phase cannot be calculated directly because there is no ‘crystal structure’ in the traditional sense. Reviewer 3 pointed out that the measured spontaneous polarization value of 4 nC/cm² differs from the calculated value of 181.93 nC/cm². Such difference is mainly due to the huge change in the arrangement of molecular dipoles after the ferroelectric crystal

is transformed into a ferroelectric liquid crystal (*Ferroelectrics*, 1989, 94, 3-62). In a solid ferroelectric crystal, the ordered molecules in the crystal structure show an orientational arrangement, where the molecular dipoles make a significant contribution to the polarization. However, in a ferroelectric liquid crystal, although the molecules are organized into planar layers, the molecules perform random rotations around the molecular long axis, so that only a fraction of the molecular dipoles make effective contribution to the polarization. In general, the polarization value of the liquid crystal phase is orders of magnitude lower than that of the solid crystal phase (*Ferroelectrics*, 1989, 94, 3-62). In addition, in the process of transforming solid crystals into liquid crystals, the volume of the 'unit cell' is also increasing, which also leads to a decrease in polarization. For the electronic polarization, considering that 4I-CASS is composed of uncharged neutral molecules, and there is only weak van der Waals interaction between molecules, we believe that there is no electronic polarization in the system.

4. How do the polarized light microscope image in Fig. 3 and the PFM image in Fig. 5 compare in the same region? Of course, it depends on the temperature, but I think we can see how the domain changes with temperature.

Response: Thank you for your good comment. The polarized light microscope images in Fig. 3 are used to determine the texture of the liquid crystal phases, while the PFM images in Fig. 5 reflect the ferroelectric domain structure of the solid phase. PFM is a scanning probe microscopy technique that cannot measure liquid crystal. Polarized light microscopy cannot characterize the ferroelectric domains in the solid phase. In addition, the liquid crystal phase has fluidity. Therefore, the domain structure information from the solid phase to the liquid crystal phase cannot be obtained *in situ*. As suggested by the reviewer, we can use PFM to characterize the temperature-dependent domain structure in the solid phases only. 4I-CASS undergoes a solid-to-solid structural phase transition at 342 K from space group $P2_1$ in phase I to space group $C2$ in phase II (Fig. 2 and 3). As shown in Fig. R1a (for review only) below, the spatial symmetry operation changes from phase I to phase II, with the appearance of 2-fold rotation axes in phase II. However, both space groups $P2_1$ and $C2$ belong to Laue point

group 2 with the same macroscopic symmetry elements of E and C_2 (Fig. R1b for review only), the spontaneous polarization direction of the crystal thus does not change (both along the b -axis) during the phase transition from phase I to phase II. According to our previous studies (*J. Am. Chem. Soc.* 2021, 143, 5091-5098), the domain structure does not change significantly for this kind of phase transition. In addition, unlike charged domain walls that the relaxation of the trapped compensation charges at high temperatures making them unstable, the domains with uncharged domain walls did not change significantly as the sample was heated (*Adv. Matter.* 2015, 27, 7832-7838). For these reasons, we believe that the domain structure in the solid phase of our sample will not change dramatically with temperature. Supplementary Fig. 20 (in revision) shows the domain structures of the two solid phases in the thin film. As expected, the domain structure did not change significantly from 303 K (phase I) to 353 K (phase II). According to your comment, we have added the corresponding figure and discussion in the revised manuscript.

Fig. R1 (for review only) | Diagram of the changes in spatial symmetry operations (a) and macroscopic symmetry elements (b) from phase I (space group $P2_1$) to phase II (space group $C2$).

The corresponding changes in the revised main text:

“We also studied the temperature-dependent evolution of the domain structures in SC

phases. As shown in Supplementary Fig. 20, the domain patterns did not change significantly from phase I to phase II. This is due to the fact that the polarization directions of the crystal have not change during the phase transition as both phases belong to point group 2.”

The corresponding changes in the revised supplementary information:

Supplementary Fig. 20 (in revision) | The domain evolution with temperature in the thin film of 4I-CASS.

REVIEWER COMMENTS

Reviewer #1 (Remarks to the Author):

The authors have answered the question we raised. We recommend to accept this manuscript.

Reviewer #2 (Remarks to the Author):

This revised manuscript is well organized, and I recommend it to be accepted.

Reviewer #3 (Remarks to the Author):

On the basis of the comments raised by me and other referees, the authors highly improved the manuscript. The main message by present experiments is clear, as Xian-Jiang Song et al. confirmed the ferroelectricity in solid and liquid phases in the same compound. I believe that the present finding is an important step for the polarization switching in liquid crystals, and highlights the importance of the ferroelectricity. Now, I would like to recommend this paper for publication in the present form.

REVIEWER COMMENTS

Reviewer #1 (Remarks to the Author):

The authors have answered the question we raised. We recommend to accept this manuscript.

Response: We sincerely thank Reviewer 1 for the excellent and valuable comments on our manuscript.

Reviewer #2 (Remarks to the Author):

This revised manuscript is well organized, and I recommend it to be accepted.

Response: We sincerely thank Reviewer 2 for spending his/her valuable time on carefully reviewing our manuscript and the in-depth comments.

Reviewer #3 (Remarks to the Author):

On the basis of the comments raised by me and other referees, the authors highly improved the manuscript. The main message by present experiments is clear, as Xian-Jiang Song et al. confirmed the ferroelectricity in solid and liquid phases in the same compound. I believe that the present finding is an important step for the polarization switching in liquid crystals, and highlights the importance of the ferroelectricity. Now, I would like to recommend this paper for publication in the present form.

Response: We sincerely thank Reviewer 3 for his/her helpful comments and questions, which are crucial for us to improve the quality of our manuscript.